An aerotaxis receptor influences invasion of Agrobacterium tumefaciens into its host

Huang Zhiwei huangzhiwei@hyit.edu.cn 1
Zou Junnan 1
Guo Minliang 2
Zhang Guoliang 1
Gao Jun 1
Zhao Hongliang 1
Yan Feiyu 1
Niu Yuan 1
Wang Guang-Long 1
1 Jiangsu Provincial Agricultural Green and Low Carbon Production Technology Engineering Research Center, School of Life Science and Food Engineering, Huaiyin Institute of Technology , Huai’an , Jiangsu Province , China
2 College of Bioscience and Biotechnology, Yangzhou University , Yangzhou City , Jiangsu Province , China
Mora-Montes Héctor
Electronic publication date: 2024 Feb 5
Publication date: 2024
Volume: 12
Electronic Location ID: e16898
Received 2023 Oct 28; Accepted 2024 Jan 16
Copyright: ©2024 Huang et al.
Copyright year: 2024
Copyright holder: Huang et al.
License: This is an open access article distributed under the terms of the Creative Commons Attribution License, which permits unrestricted use, distribution, reproduction and adaptation in any medium and for any purpose provided that it is properly attributed. For attribution, the original author(s), title, publication source (PeerJ) and either DOI or URL of the article must be cited.
License URL: https://creativecommons.org/licenses/by/4.0/

Keywords: Agrobacterium tumefaciens, Chemotaxis, Aerotaxis, Chemoreceptor, Biofilm formation, Pathogenicity

Funding: the National Natural Science Foundation of China 32000051 31870118 31771736 The Natural Science Foundation of Jiangsu Province No. BK20201068 The Key Projects of Key Research and Development Programs of Jiangsu Province No. BE2021323 The Talent Introduction Research Project of Huaiyin Institute of Technology No. Z301B19536 This work was supported by the National Natural Science Foundation of China (No. 32000051, 31870118, 31771736), the Natural Science Foundation of Jiangsu Province (No. BK20201068), the Key Projects of Key Research and Development Programs of Jiangsu Province (No. BE2021323), and the Talent Introduction Research Project of Huaiyin Institute of Technology (No. Z301B19536). The funders had no role in study design, data collection and analysis, decision to publish, or preparation of the manuscript.

==============================
Agrobacterium tumefaciens is a soil-borne pathogenic bacterium that causes crown gall disease in many plants. Chemotaxis offers A. tumefaciens the ability to find its host and establish infection. Being an aerobic bacterium, A. tumefaciens possesses one chemotaxis system with multiple potential chemoreceptors. Chemoreceptors play an important role in perceiving and responding to environmental signals. However, the studies of chemoreceptors in A. tumefaciens remain relatively restricted. Here, we characterized a cytoplasmic chemoreceptor of A. tumefaciens C58 that contains an N-terminal globin domain. The chemoreceptor was designated as Atu1027. The deletion of Atu1027 not only eliminated the aerotactic response of A. tumefaciens to atmospheric air but also resulted in a weakened chemotactic response to multiple carbon sources. Subsequent site-directed mutagenesis and phenotypic analysis showed that the conserved residue His100 in Atu1027 is essential for the globin domain’s function in both chemotaxis and aerotaxis. Furthermore, deleting Atu1027 impaired the biofilm formation and pathogenicity of A. tumefaciens. Collectively, our findings demonstrated that Atu1027 functions as an aerotaxis receptor that affects agrobacterial chemotaxis and the invasion of A. tumefaciens into its host.

Introduction

Agrobacterium tumefaciens (reclassified as Agrobacterium fabrum) is a soil-borne phytopathogen that causes crown gall disease in a wide range of dicotyledonous plants and some gymnosperms (Gelvin, 2012; Păcurar et al., 2011). It can transfer its T-DNA (transferred DNA) into the host plant’s genome, and genetically transform the hosts (Gelvin, 2017; Guo et al., 2019). The extraordinary capacity of horizontal gene transfer has made A. tumefaciens a powerful and widely used tool for obtaining transgenic or gene-edited plants (Guo et al., 2019; Huang et al., 2018). A. tumefaciens is also a model microorganism to study the interaction between microbes and plant hosts, or microbes (Brown et al., 2023; Lang & Faure, 2014). A. tumefaciens usually distributes around the plant rhizosphere. They specifically sense and recognize some signal molecules, such as phenolic compounds (acetosyringone, hydroxy-acetosyringone) and sugar compounds, released by wounded plant tissue (Păcurar et al., 2011). Chemotaxis offers A. tumefaciens cells the ability to track these environmental cues and move toward the plants, which is crucial for A. tumefaciens to find and infect host plants. In the last four decades, the molecular mechanism of the pathogenicity of A. tumefaciens and horizontal gene transfer from A. tumefaciens to host plants has been well understood (reviewed in Bourras, Rouxel & Meyer, 2015; Gelvin, 2017; Guo et al., 2019; Matveeva & Lutova, 2014). At the same time, research on the molecular mechanism of A. tumefaciens chemotaxis has lagged.

As a typical behavior of bacteria, chemotaxis has been well-studied in Escherichia coli (Bi & Sourjik, 2018; Flack & Parkinson, 2022; Parkinson, Hazelbauer & Falke, 2015). The chemotaxis system of E. coli contains five chemoreceptors and six core components: CheA, CheW, CheY, CheB, CheR, and CheZ. Two chemoreceptor trimers of dimers, a histidine autokinase CheA dimer, and two coupling proteins CheW form a ternary signaling complex at the cell poles (Parkinson, Hazelbauer & Falke, 2015). In the ternary signaling complex, chemoreceptors perceive and bind to diverse signal molecules through the utilization of their ligand-binding domains (LBDs) (Bi & Sourjik, 2018). CheW couples the autophosphorylation activity of CheA to chemoreceptor control (Flack & Parkinson, 2022). The phosphorylated CheA transfers its phosphoryl groups to the response regulator protein CheY and the methylesterase CheB. The phosphorylated CheY is accountable for eliciting alterations in the flagellar rotation direction, transitioning from counterclockwise (CCW) to clockwise (CW) (Huang et al., 2018). On the other hand, the phosphorylated CheB, in collaboration with the methyltransferase CheR, is employed to fine-tune the chemoreceptors’ sensitivity to external stimuli (Parkinson, Hazelbauer & Falke, 2015; Salah Ud-Din & Roujeinikova, 2017). Upon binding to chemoattractants, the ternary signal complex undergoes conformational changes, leading to the inhibition of autophosphorylation of CheA. This inhibition results in a decrease in the flow of CheA phosphoryl groups to CheY and CheB. The phosphorylated form of CheY is also susceptible to dephosphorylation by its associated phosphatase CheZ (Parkinson, Hazelbauer & Falke, 2015). Consequently, the short half-life of phosphorylated CheY enables the bacterial cells to respond to chemotactic stimuli in a unidirectional manner. Simultaneously, the reduction in phosphorylated CheB allows CheR to continue methylating chemoreceptors. Methylation increases chemoreceptors’ ability to activate CheA, resulting in adaptation (Parkinson, Hazelbauer & Falke, 2015; Salah Ud-Din & Roujeinikova, 2017).

In bacteria, chemoreceptors can recognize a variety of chemoeffectors. Chemoeffectors encompass a range of chemical compounds or salts, such as sugars (Day et al., 2016; Ye et al., 2021), amino acids (Kumar Verma, Samal & Chatterjee, 2018; Ye et al., 2021), indole-3-acetic acid (Rico-Jimenez et al., 2022), and metal ions (Shi et al., 2017). Additionally, certain environmental factors, such as light (Perlova et al., 2019), redox (Xie et al., 2010), pH (Sweeney et al., 2012), and oxygen (Bibikov et al., 1997; Garcia et al., 2017). Oxygen plays a crucial role in the survival of aerobic and microaerophilic microorganisms (Zhulin et al., 1996). Aerotaxis, a process reliant on chemoreceptors to detect oxygen or redox alterations, enables microorganisms to navigate towards areas with optimal oxygen concentrations, facilitating various biological processes such as energy generation and nitrogen fixation (Jiang et al., 2016; Zhulin et al., 1996). In the last three decades, aerotaxis has attracted the attention of many researchers and several oxygen-related chemoreceptors have been identified in bacteria and archaea (Arrebola & Cazorla, 2020; Bibikov et al., 1997; Freitas et al., 2005; Garcia et al., 2017; Hou et al., 2000; Hou et al., 2001a; Hou et al., 2001b; Jiang et al., 2016; Tumewu et al., 2022).

The ligand-sensing specificity of chemoreceptors is affected by the type of LBDs. Two types of LBDs within chemoreceptors have been proven to be involved in aerotaxis (Edwards, Johnson & Taylor, 2006; Freitas et al., 2005; Hou et al., 2000; Xie et al., 2010). The first LBD, known as the Per-Arnt-Sim (PAS) domain, serves a multifaceted role. Certain chemoreceptors that possess the PAS domain function as redox sensors, monitoring alterations in the redox status of the electron transport system through the flavin adenine dinucleotide (FAD) cofactor associated with their PAS domains (Edwards, Johnson & Taylor, 2006; Xie et al., 2010). This enables the cells to actively seek out favorable oxygen conditions. On the other hand, other chemoreceptors employ their PAS domains to directly sense oxygen through a heme-bound cofactor, thereby regulating aerotaxis (Jiang et al., 2016; Garcia et al., 2017; Tumewu et al., 2022). The second LBD corresponds to the globin domain, and chemoreceptors containing this domain serve as direct oxygen sensors reliant on heme cofactor (Freitas et al., 2005; Hou et al., 2000). In addition, it has been observed that certain chemoreceptors, characterized by possessing 4-helix-bundle (4HB)-type LBDs, exhibit responsiveness to variations in proton motive force in aerotaxis, exemplified by Tsr in E. coli (Edwards, Johnson & Taylor, 2006). However, the specific involvement of 4HB-type LBDs in aerotaxis remains uncertain. Moreover, aside from their involvement in chemotaxis and aerotaxis, the identified oxygen-related chemoreceptors also play a role in the interactions between microorganisms and their hosts (Arrebola & Cazorla, 2020; Garcia et al., 2017; Jiang et al., 2016; Tumewu et al., 2022). Consequently, investigating oxygen-related chemoreceptors will provide insights into the physiological mechanisms of microorganisms and their interactions with hosts.

A. tumefaciens strains possess numerous genes that are predicted to encode chemoreceptors (Shi et al., 2017; Wood et al., 2001; Xu et al., 2020). Large numbers of chemoreceptors in A. tumefaciens suggest that the bacteria can detect diverse chemical molecules or environmental stimuli to navigate the way to favorable ecological niches (Xu et al., 2020). To date, the identification of chemoreceptors in A. tumefaciens has been limited to only four (Shi et al., 2017; Wang et al., 2021; Ye et al., 2021; Zong et al., 2022). In the case of A. tumefaciens C58, Atu0514, which is predicted to possess a globin domain as its LBD, has been implicated in the chemotactic response to sucrose, valine, citric acid, and acetosyringone (AS) (Ye et al., 2021). Atu0526 has been found to use a single-Cache domain as LBD to bind the broad antibacterial agent formic acid (Wang et al., 2021). Additionally, Atu1912, another chemoreceptor featuring a single-Cache domain, has been shown to mediate the attractant response to pyruvate and the repellent response to propionic acid. The ligands of the single-Cache domain within Atu1912 are pyruvate and propionic acid (Zong et al., 2022). In another strain, A. tumefaciens GW4, a putative chemoreceptor harbors undefined LBD not only regulated the chemotactic response towards trivalent arsenic, but also affected trivalent arsenic oxidation, trivalent arsenic resistance, and bacterial growth (Shi et al., 2017). However, chemoreceptors related to oxygen have not yet been identified in A. tumefaciens.

In this study, we characterized a globin domain-containing chemoreceptor in A. tumefaciens C58 named Atu1027 and showed that Atu1027 is a globin-coupled sensor that modulates aerotaxis and chemotaxis. We also analyze the impact of Atu1027 on biofilm formation and pathogenicity of A. tumefaciens.

Materials & Methods

Bacterial strains, plasmids, media, and growth conditions

All Escherichia coli and Agrobacterium tumefaciens strains and plasmids used in this study are described in Table S1. E. coli strains were grown in the lysogeny broth (LB) medium with or without 1.5% agar at 37 °C or 25 °C. A. tumefaciens C58 and its derivatives were grown in MG/L medium or AB-sucrose medium at 28 °C (Cangelosi et al., 1991; Gelvin, 2006). The final concentrations of antibiotics: for E. coli, ampicillin (Ap) at 100 µg/mL, kanamycin (Km) at 50 µg/mL, carbenicillin (Cr) at 25 µg/mL, tetracycline (Tc) at 15 µg/mL, chloramphenicol (Cm) at 34 µg/mL; for A. tumefaciens, kanamycin (Km) at 100 µg/mL.

In silico sequence analysis and phylogenetic analysis

All sequences utilized in the current investigation were acquired from the National Center for Biotechnology Information (NCBI) database. Likewise, a protein BLAST analysis was employed for the examination of the sequences. The RefSeq Select proteins database was utilized to search for proteins with homology. The Pfam database was utilized to analyze the sequence of Atu1027. Amino acid sequence alignments were conducted using DNAman8 or ClusterW software. The transmembrane sequence of Atu1027 was analyzed using the online tools DeepTMHMM (https://dtu.biolib.com/DeepTMHMM) and ProtScale (https://web.expasy.org/protscale/). Evolutionary analyses were conducted in MEGA11 (Tamura et al., 2021). The evolutionary history was inferred using the Neighbor-Joining method (Saitou & Nei, 1987). The bootstrap consensus tree inferred from 1,000 replicates was taken to represent the evolutionary history of the taxa analyzed (Felsenstein, 1985). Branches corresponding to partitions reproduced in less than 50% bootstrap replicates were collapsed. The percentage of replicate trees in which the associated taxa clustered together in the bootstrap test (1,000 replicates) was shown next to the branches (Felsenstein, 1985). The evolutionary distances were computed using the JTT matrix-based method (Jones, Taylor & Thornton, 1992) and were in the units of the number of amino acid substitutions per site. This analysis involved 49 amino acid sequences. All positions with less than 50% site coverage were eliminated. There was a total of 502 positions in the final dataset.

DNA manipulations

Molecular manipulations of DNA were conducted according to standard molecular procedures. All primers used in this study are described in Table S2. Purification of PCR products and DNA fragments from agarose gels was performed by using FastPure Gel DNA Extraction Mini Kit (Vazyme Biotech, Nanjing, China). Plasmid DNA was purified with the TIANprep Mini Plasmids Kit (Tiangen Biotech, Beijing, China). According to the Inoue protocol, the competent cells of E. coli were prepared (Inoue, Nojima & Okayama, 1990). The plasmid DNA was transferred by heat shock into E. coli competent cells (Sambrook, Fritsch & Maniatis, 1989). The competent cells of A. tumefaciens were prepared as previously described (Cangelosi et al., 1991). All plasmids were electrotransformed into A. tumefaciens strains using an Eppendorf Eporator (Eppendorf AG, Hamburg, Germany).

Construction of deletion, complementation, or point mutants of A. tumefaciens

Following the principle of homologous recombination, the precise deletion of the atu1027 gene in A. tumefaciens C58 was constructed by the procedures as previously described (Guo, Zhu & Gao, 2009). The gene-deletion strategy was shown in Fig. S1. Briefly, a fusion of 585-bp upstream and 587-bp downstream of the atu1027 open reading frame (ORF) was inserted into the suicide plasmid pEX18Km via the Bam H I and Eco R I restriction sites, resulting in the creation of pEX-d1027. In pEX18Km and its derived plasmids, the kanamycin-resistant gene nptIII was used as a positive selection marker, the suicide gene sacB was used as a counterselection marker (Huang et al., 2018). Subsequently, pEX-d1027 was introduced into A. tumefaciens C58. Two rounds of selection were conducted to identify potential gene-deficient mutants. The colonies were further subjected to PCR screening and DNA sequencing for verification purposes. The avirulent mutant C58 ΔvirA was constructed using the same strategy. The complemented strain of the atu1027-deficient mutant was implemented by the introduction of the plasmid pCB301-1027. The plasmid pCB301-1027 carries a 479-bp promoter region and a 1500-bp intact ORF of the atu1027 gene.

To substitute the conserved histidine residue at position 100 within the globin region of Atu1027 with alanine, two single-residue mutation primers (cp1027-H100A-2 and cp1027-H100A-3) were designed (Table S2). Two primers for amplifying the atu1027 expression cassette, in conjunction with the two single-residue mutation primers, were utilized to procure two DNA fragments. The specific site of the two base mutations is situated within the region of overlap between the two aforementioned DNA fragments. Using overlap PCR, the atu1027 expression cassette containing the two base mutation sites was generated and subsequently inserted into the plasmid pCB301 via the Bam H I and Eco R I restriction sites. The recombinant plasmid pCB-1027H100A was then introduced into the atu1027-deficient mutant via electroporation.

Bacterial two-hybrid assay

The bacterial two-hybrid system (Stratagene, Agilent Technologies Inc., Santa Clara, CA, USA) was used to detect potential protein–protein interactions. The ORF of atu1027 was amplified and inserted into the bait plasmid pBT to express λcI-Atu1027 fusion protein. The ORFs of cheW1 and cheW2 were amplified and inserted into the target plasmid pTRG to express CheW1–RNAP and CheW2–RNAP fusion proteins, respectively. Then the recombinant target plasmid and bait plasmid were collectively transferred into E. coli XL1-Blue MR competent cells by heat shock and spread on LB-CTCK plates for screening positive clones. The LB-CTCK medium is made up of LB medium, 25 µg/mL carbenicillin, 15 µg/mL tetracycline, 34 µg/mL chloramphenicol, and 50 µg/mL kanamycin. The positive clones in the LB-CTCK plates were picked out and streaked onto a new LB-TCK plate that contains 15 µg/mL tetracycline, 34 µg/mL chloramphenicol, 50 µg/mL kanamycin, 80 µg/mL X-gal, and 0.2 mM galactosidase inhibitor. The LB-TCK plate with positive clones was placed in a lightless biochemical incubator at 37 °C for 16 h. The galactosidase activity of different groups was quantified as previously described (Ye et al., 2021).

Protein expression, pull-down assay, and immunoblotting

E. coli BL21 (DE3) was employed as the host organism for the overproduction of Atu1027-His, Atu1027H100A-His, CheW1, CheW2. E. coli cells containing pET-30a and its derived recombined plasmids were cultivated until reaching a cell density of approximately 5 × 108 cell/mL. Subsequently, isopropylthio-β-D-galactoside (IPTG) was added to the cultures at a final concentration of 0.2 mM to induce the expression of the fusion proteins. After being cultivated at 25 °C for 6 h, the cells were centrifuged 5,000 × g for 10 min, washed twice and resuspended in PBS buffer (10 mM, pH 7.4). The pull-down assay was conducted following the methodology described in a previous study (Huang et al., 2018). The cell suspension was subjected to sonication until it reached a state of near clarity, and the supernatant from this sonicated cell suspension was collected through centrifugation at a force of 12,000 × g for a duration of 30 min. Equal amounts of supernatant that from different BL21(DE3) cell lysates were then respectively introduced into one mL Ni-IDA Resins (Sangon Biotech, Shanghai, China). The supernatant obtained from the lysates of BL21(DE3) containing pET30a was used as the negative control. The resins were subjected to five washes with PBS buffer and subsequently incubated with two mL of cell crude extract derived from a 50 mL cell culture that overproduced either CheW1 or CheW2. Following a 1-hour incubation at 4 °C with gentle shaking, the resins were washed with 10 volumes of PBS buffer containing increasing concentrations of imidazole (10 mM, 30 mM, and 50 mM) to remove nonspecifically bound proteins. The specifically bound proteins were then eluted from the resins using 500 µL of PBS buffer containing 250 mM imidazole. 5 µL of eluent obtained from distinct samples were subjected to SDS-PAGE. For immunoblotting, proteins were transferred onto hydrophilic polyvinylidene difluoride membranes (Merck Millipore, Billerica, MA, USA) through electrophoresis. The visualization of the transferred proteins was achieved using the BCIP/NBT alkaline phosphatase color development kit (Beyotime Biotechnology, Shanghai, China), following the recommended protocol provided by the manufacturer. To obtain antibodies capable of distinguishing between two CheWs, peptide fragments corresponding to the variable sequences of CheW1 (amino acid residues 142-155) and CheW2 (amino acid residues 146-159) were synthesized artificially and utilized as antigens for antibody generation in New Zealand rabbits (Huang et al., 2018).

Air trap assay for aerotaxis

The procedure for the air trap assay was followed as described by Arrebola & Cazorla (2020) with some modifications (Arrebola & Cazorla, 2020). To establish an air trap, a modified Pasteur pipette was initially placed within a glass tube, covered with a lid, and subjected to autoclaving. Next, sterile AB-sucrose medium containing 0.1% agar was heated and carefully poured into the glass tube. The liquid level of the medium was maintained three cm below the top of the Pasteur pipette. Following this, two mL of paraffin oil was added to the glass tube but was prevented from falling into the Pasteur pipette. Consequently, the paraffin oil acted as a barrier, effectively isolating the external medium of the Pasteur pipette from the surrounding air and creating a localized hypoxic environment. Only the medium contained within the Pasteur pipette was able to interact with the air and undergo air exchange. Subsequently, A. tumefaciens strains that grew to the middle-logarithmic growth phase were gently centrifuged at 3,500 × g for 3 min and resuspended in fresh AB-sucrose medium to achieve an OD600 of 0.6. 100 µL of bacterial cells were slowly injected into AB-sucrose medium with a disposable sterilized syringe but outside of the Pasteur pipette. The inoculation site was positioned two cm above the base of the Pasteur pipette. If the bacteria possess oxygen-sensing capabilities, they would exhibit movement toward the interior of the Pasteur pipette. Finally, the air trap devices were placed in a biochemical incubator without any disturbance and observed every 12 h, until the bacterial population was visible.

Aerotaxis behavior observation

The aerotaxis behavior observation was conducted as previously described with some modifications (Arrebola & Cazorla, 2020). A. tumefaciens strains were grown in AB-sucrose medium to middle-logarithmic growth phase, centrifuged at 3,500 × g for 3 min, and suspended at the same cell density (1 ×108 cfu/mL) in chemotaxis buffer (10 mmol/L KH2PO4, 0.1 mmol/L EDTA, pH 7.0) with 14.6 mM sucrose as the energy source. 3 µL drop of bacterial cells was deposited on a coverslip, the coverslip was inverted on an excavated slide. As a result, the drop was surrounded by air. The bacterial behavior was monitored by optical microscopy at 1,000 × magnification. The photographs of bacterial behavior at the edges of the droplets were taken after 5, 10, and 15 min. To prevent bacterial motility from being orientated by the light during the photo interval, the microscope light was turned off.

Chemotaxis assays

Chemotaxis to different carbon sources in a spatial gradient was assayed in the soft agar plates containing AB-medium, 0.2% Bacto agar (Beyotime Biotech, Shanghai, China), and 15 mM single carbon source. The carbon sources included mannitol, xylose, fructose, sucrose, glucose, and galactose. A. tumefaciens cells were grown and collected as indicated above for aerotaxis. The tested cell cultures were normalized to an OD600 of 0.6. 3 µL of tested cell suspension was inoculated into the soft agar plates and incubated for 36 h without disturbance. The diameter of the swimming halo was measured and used to quantify the chemotaxis to different carbon sources.

Biofilm formation assays

A. tumefaciens cells in the middle-logarithmic growth phase were collected and normalized to an OD600 of 0.1. two mL of culture was added to a sterile glass tube. The tubes were incubated for 72 h. For crystal violet (CV) visualization, the tube was washed thoroughly with double-distilled H2O and stained with 0.15% (wt/vol) crystal violet staining solution for 15 min. After the staining procedure, the tubes were washed five times with double-distilled H2O to remove the crystal violet staining solution. The formation of purple ring-like structures on the tube was observed. Finally, the crystal violet is eluted off with 30% acetic acid (vol/vol). The eluent is measured at 570 nm to determine the strength of biofilm formation. A normalization of absorbance values to culture growth was achieved by dividing the solubilized CV ODnm value by the planktonic cell OD600 value. All strains were performed in three replicates.

Tumorigenesis assays

The tumorigenesis assays on the leaves of the Kalanchoe plant were implemented as described previously with some modifications (Yang et al., 2020). Different A. tumefaciens strains were prepared as in the air trap assay described previously. An avirulent strain C58 ΔvirA was used as a negative control. The agrobacterial cells were resuspended in AB-sucrose liquid medium at 5 ×108 cfu/mL for inoculation. Kalanchoe plants’ leaves that grew for more than three weeks were selected for the infection. Before inoculation, several wound lines were made in the Kalanchoe leaf by using a sterile scalpel. Each wound line was inoculated with 3 µL of different cell suspensions. The infected Kalanchoe plants were grown in the pots at room temperature for 30–40 days. The tumours in the wound lines were photographed and weighed. Each inoculation was repeated at least three times on separate leaves.

Statistical analysis

The data were presented as the mean ± standard error of three replicates using analysis of variance (ANOVA). To assess the significance of differences between various groups, the Student-Newman-Keuls (SNK) multiple comparison test was employed at a significance level of P < 0.05.

Results

Atu1027 is a cytoplasmic chemoreceptor containing a globin domain

The globin domain of the heme-containing transducers (HemAT-Bs from Bacillus subtilis and HemAT-Hs from Halobacterium salinarum) can bind diatomic oxygen and mediate aerotactic reactions in bacteria and archaea (Hou et al., 2000; Hou et al., 2001a). A. tumefaciens C58 has 20 genes coding putative chemoreceptors on its genome (Wood et al., 2001). Among these genes, atu1027, which is located on the circular chromosome, is predicted to encode a chemoreceptor (accession number: WP_010971342.1) with a globin domain (Fig. 1A). The Pfam database revealed that the peptide chain of the globin domain within Atu1027 encompasses 159 amino acid residues. A prior investigation proposed that the globin domains of Atu1027, HemAT-Bs, and HemAT-Hs evolved from a shared ancestral globin (Freitas et al., 2005). In the present study, amino acid sequence alignment (Fig. 1B) indicated that the globin domain of Atu1027 shares 17.65% sequence identity with that of HemAT-Bs and 18.64% sequence identity with that of HemAT-Hs. The alignment of the sequence also demonstrated that the globin domain of Atu1027 retains two strictly conserved residues present in all globin domains (Hou et al., 2001a). In addition, the analysis conducted using DeepTMHMM and ProtScale indicated that Atu1027 does not possess a transmembrane domain. These findings suggested that Atu1027 potentially functions as a cytoplasmic chemoreceptor involved in mediating aerotaxis.

Figure 1 Sequences and domain information of Atu1027.

(A) Domains of HemAT-Bs (Bacillus subtilis), HemAT-Hs (Halobacterium salinarum), and Atu1027 (Agrobacterium tumefaciens) predicted by Pfam and PROSITE servers. (B) Amino acid sequence alignment of globin domain of HemAT-Bs, HemAT-Hs, and Atu1027. The amino acid full alignment was created using the Clustal program of DNAman8 software. All settings used default values. The numbers on the right of the figure indicate the position of the amino acid. Asterisks signify the residues that are conserved across all globin domains, whereas the dark spots indicate residues that are highly conserved in all globin domains (Bashford, Chothia & Lesk, 1987; Vinogradov, Walz & Pohajdak, 1992; Hou et al., 2000).

The initial comprehensive alignment of the complete sequence indicated a significant degree of conservation of Atu1027 across various Agrobacterium species (Fig. S2). Consequently, a more extensive alignment of the entire sequence and subsequent construction of a phylogenetic tree were undertaken to investigate the evolutionary lineage of Atu1027. A total of 48 sequences, exhibiting a sequence identity of over 50% with Atu1027, were carefully selected for the construction of the phylogenetic tree. Figure 2 demonstrates the division of 49 proteins from 11 bacterial genera into three distinct groups. Notably, the clustering of proteins within the same bacterial genus is readily apparent. These proteins primarily originate from Agrobacterium, Shinella, Neorhizobium, and Ciceribacter. However, the proteins derived from Rhizobium in the tree were not effectively clustered. Notably, the protein WP 082184016.1 of Rhizobium oryzihabitans and the protein WP 045017035.1 of Rhizobium nepotum were grouped together in group 2, alongside Atu1027 and five other proteins from Agrobacterium. Furthermore, the protein comparison revealed that WP 082184016.1 and WP 045017035.1 exhibit a higher degree of similarity with Atu1027. This suggests that these two proteins may potentially possess similar functional characteristics to Atu1027.

Figure 2 Consensus tree of phylogenetic relationships of Atu1027 and its homologs.

The alignment of the proteins was carried out by Clustal W, and the phylogenetic tree was established using MEGA 11 by the neighbor-joining (NJ) method with 1,000 bootstrap replicates. The yellow font represents Atu1027 protein. The red triangles represent bootstraps (range: 0.512–1). For information about the sequences used in the consensus phylogenetic tree, see Table S3.

Atu1027 interacts with CheW1 and CheW2

In the chemotaxis system of E. coli, the formation of the ternary signaling complex relies on the interaction of chemoreceptors and the unique coupling protein CheW (Parkinson, Hazelbauer & Falke, 2015). A. tumefaciens C58 two coupling proteins, namely CheW1 and CheW2 (Huang et al., 2018; Wood et al., 2001). In light of the hypothesis that Atu1027 serves as a cytoplasmic chemoreceptor, it is crucial to validate the potential interaction between Atu1027 and two CheWs. To assess the protein-protein interactions, the bacterial two-hybrid and in vitro His-tag pull-down assays were employed. In the bacterial two-hybrid assay, the presence of blue colonies on the designated plates serves as an indication of the interaction between the target protein and the bait protein. The intensity of the blue color corresponds to the strength of the protein-protein interaction and the level of β-galactosidase activity. As illustrated in Figs. 3A and 3B, the experimental groups (CheW1+Atu1027 and CheW2+Atu1027) displayed a higher β-galactosidase activity compared to the negative control group. In the His-tag pull-down assay (Fig. 3C), both CheW1 and CheW2 were identified in the eluate from the resin bound with Atu1027-His, while they were absent in the negative control. This outcome serves as confirmation that both CheW1 and CheW2 can interact with Atu1027 in an in vitro setting. Consequently, the results indicated that Atu1027 interacts with either CheW1 or CheW2.

Figure 3 The interactions between Atu1027 and two CheWs.

(A) Bacterial two-hybrid assay. The E. coli cells co-expressing four combinations of target protein and bait protein were streaked on the X-gal indicator plate. The first combination is a positive control expressing LGF2 and Gal11, two proteins that are known to interact with each other; the second combination expresses Atu1027 and CheW1; the third combination expresses Atu1027 and CheW2; the fourth combination is a negative control expressing a single LGF2 protein. The interaction of the target protein and bait protein will activate the activity of β-galactosidase to catalyze the hydrolysis of X-gal, leading to the appearance of blue colonies. The chromogenic reaction of colonies harboring different combinations of target protein and bait protein are shown on the X-gal indicator plate. (B) Quantification of β-galactosidase activity of different combinations. The data represents the means ± standard deviations from three separate experiments, each with triplicate samples. Significant differences between groups are indicated by different lowercase letters on the bar (P < 0.05). (C) The interactions between Atu1027 and two CheWs were tested using a His-tag pull-down assay. The supernatants, both with and without overproduced Atu1027-His, were separately bound to the Ni-IDA resins and subsequently rinsed with PBS buffer. This was followed by the pull-down of overproduced CheW1 or CheW2. The supernatant lacking overproduced Atu1027-His was employed as a negative control to validate the selectivity of Atu1027-His binding to CheWs.The original western blot image can be found in Fig. S3.

Atu1027 is an aerotaxis receptor in A. tumefaciens C58

To ascertain the functions of Atu1027, a precise deletion of atu1027 in A. tumefaciens C58 was accomplished via homologous recombination, as outlined in the Materials and Methods section. The resulting atu1027-deficient mutant was denoted as C58 Δ1027. Subsequently, the atu1027 gene, along with its native promoter (located 479 bp upstream of atu1027), was reintroduced into the C58 Δ1027 mutant using a plasmid, thereby restoring the expression of Atu1027 protein in C58 Δ1027. This complemented strain was designated as C58 Δ1027+. The bioinformatic analysis and protein interaction experiments provided indications that Atu1027 could potentially function as an aerotaxis receptor in A. tumefaciens. To validate this hypothesis, the aerotaxis of various A. tumefaciens strains was assessed using air trap assays and optical microscope observations.

The results of the air trap assays are presented in Fig. 4. At 72 h post-inoculation, the device containing the complemented strain C58 Δ1027+ exhibited a significant aggregation of bacterial cells on the surface of the semi-solid medium in the Pasteur pipette. In contrast, no such phenomenon was observed in the device containing the wild-type C58 or the atu1027-deficient mutant C58 Δ1027. At 96 h post-inoculation, the wild-type C58 cells displayed visible aggregation on the surface of the semi-solid medium, while the medium inoculated with C58 Δ1027 remained clear in the Pasteur pipette. At 120 h post-inoculation, the medium inoculated with C58 Δ1027 remained clear on the surface of the medium in the Pasteur pipette. The results above indicated that the deletion of atu1027 eliminates the aerotactic response of A. tumefaciens to atmospheric air. Furthermore, we conducted growth curve experiments in AB-sucrose liquid medium to determine if the deletion of atu1027 affects the growth rate of A. tumefaciens. The growth rates of C58 Δ1027 and C58 Δ1027+ were comparable to that of the wild-type C58 (Fig. S4), and the difference was not statistically significant (P value > 0.05). Hence, the deletion of atu1027 did not affect the normal growth of A. tumefaciens.

Figure 4 Air trap assays of wild-type C58, atu1027-deficient mutant C58Δ1027, and its complemented strain C58Δ1027+.

(A) A schematic of the air trap device. The device consists of a glass tube filled with AB-sucrose semi-solid medium, a Pasteur pipette, a rubber ring, and a lid made of aluminum foil. The glass tube was sterilized by autoclave with the Pasteur pipette. A rubber ring was tied to the upper end of the Pasteur pipette to keep it upright in the glass tube. To facilitate air diffusion and exchange, the Pasteur pipette’s mouth was shaped into a pointed tip. 2 mL of the liquid paraffin was gently added to the medium, but not the Pasteur pipette. (B) Pictures represent bacterial behavior in the air trap device. If the air is detected by A. tumefaciens and acts as an attractant, the A. tumefaciens cells will move to the medium surface and gather at the favorable position inside the Pasteur pipette. The yellow arrows mark the position of the bacterial cells after incubation.

Subsequently, an optical microscope was employed to perform continuous monitoring of the aerotaxis exhibited by the tested agrobacterial strains. The bacterial cells were deposited onto a coverslip, which was promptly inverted onto an excavated slide. To prevent air infiltration, Vaseline was applied to all four sides of the coverslip. The slide was then positioned on the objective table of the optical microscope and observed at 5-minute intervals. The findings are illustrated in Fig. 5. At 5 min post-inoculation, C58 and C58 Δ1027+ cells began to accumulate at the boundary of bacterial suspension and air. The quantity of C58 and C58 Δ1027+ cells was significantly higher compared to that of C58 Δ1027. At 10 and 15 min post-inoculation, there was further accumulation of C58 and C58 △1027+ cells at the boundary between the bacterial suspension and the air, while the number of C58 △1027 cells did not show a significant increase. These results are consistent with the results obtained from the air trap assays. Collectively, it can be concluded that Atu1027 is an aerotaxis receptor that modulates in the aerotaxis of A. tumefaciens under the test condition.

Figure 5 Bacterial aerotaxis behavior observation under the optical microscope.

In the experiment, middle-logarithmic A. tumefaciens strains were washed and suspended in a chemotaxis buffer with 14.6 mM sucrose for the energy supply. The coverslips were inverted over excavated slides after 2 µL of bacterial cells were deposited on the coverslip. This resulted in the air surrounding the drop. Then A. tumefaciens strains were observed under optical microscopy at 1000 × magnification. 5, 10, and 15 min after the droplets were dropped, photographs were taken of bacterial behavior at the edges of the droplets. Videos showing the bacterial response to atmospheric air were also provided in the supplementary material. Video S1: wild type C58; Video S2: C58Δ1027; Video S3: C58Δ1027+.

Atu1027 is involved in the chemotaxis of A. tumefaciens

In A. tumefaciens C58, CheW1 and CheW2 are involved in the formation of ternary signaling complexes at the cell poles (Huang et al., 2018). Atu1027 interacts with both CheW1 and CheW2, indicating its role as a component of the chemotaxis system in A. tumefaciens. The deletion of the atu1027 gene may have an impact on the chemotactic abilities of A. tumefaciens. Therefore, in order to assess the chemotaxis of A. tumefaciens, six common carbon sources (mannitol, xylose, fructose, sucrose, glucose, galactose) were tested on the swim agar plates. As shown in Fig. 6, in six different kinds of semi-solid AB-medium containing different sugars, the swimming halo of the atu1027-deficient mutant C58 Δ1027 is slightly smaller than that of wild-type C58, demonstrating that the deletion of atu1027 slightly reduced the chemotactic response to multiple carbon sources in A. tumefaciens. Additionally, when the atu1027 gene was re-introduced into the atu1027-deficient mutant C58 Δ1027, the swimming halo of the complemented strain C58 Δ1027+ was bigger than that of wild-type C58. This may be caused by an increased copy number of the atu1027 gene in the complemental strain C58 Δ1027+.

Figure 6 Comparison of chemotactic behavior between wild-type C58, atu1027-deficient mutant C58Δ1027, and its complemented strain C58Δ1027+.

Equal amounts of cells from different A. tumefaciens strains that were grown to the middle-log phase were inoculated on the swimming plates. The diameter of the bacterial colony after growing for 48 h on the swimming plate was used to quantify the chemotactic response of A. tumefaciens to six common carbon sources. (A) Typical colonies of different A. tumefaciens strains at 48 h post-inoculation on the swimming plate. (B) Statistical results of swimming halo diameter from different A. tumefaciens strains at 48 h post-inoculation on the swimming plate. The data are the means from four plates with standard errors. The lowercase letters above the bars represent significant differences obtained by one-way analysis of variance, followed by all pairwise multiple comparison procedures (SNK test) p < 0.05 (DPS software). C58, A. tumefaciens wild-type C58; C58Δ1027, atu1027-deficient mutant that derives from C58; C58Δ1027+, the complementation strain of C58Δ1027 carrying pCB301-C1027; C58Δw, cheW1_-cheW2_ double-deficient mutant deriving from C58 (as the chemotaxis-negative strain) (Huang et al., 2018).

The replacement of His100 with alanine completely abolished the function of the globin domain of Atu1027

In the globin domains of HemAT-Bs and HemAT, the proximal histidine residue (His-123) plays a crucial role in both oxygen sensing and heme binding (Freitas et al., 2005; Hou et al., 2000; Hou et al., 2001a). Replacing the His-123 with alanine eliminates the function of the globin domains (Hou et al., 2001a). The globin domain of Atu1027 contains a histidine residue at position 100 (His100), which has been predicted to serve as the heme binding site (Fig. 1). Mutation on His100 may also affect the function of Atu1027. Therefore, to assess the importance of His100 for the function of Atu1027, a site-directed mutagenesis technique was employed to replace His100 with an alanine residue. The H100A mutant of Atu1027 was expressed in the C58 Δ1027 strain, resulting in the creation of the mutant strain C58 Δ1027H100A. Chemotaxis assay and air trap assay for aerotaxis were conducted to test the phenotype of the mutant strain C58 Δ1027H100A. As anticipated, the C58 Δ1027H100A strain exhibited a compromised chemotactic response to sucrose (Figs. 7A and 7B) and did not accumulate in the Pasteur pipette after 120 h of inoculation (Fig. 7C). These phenotypic characteristics of C58 Δ1027H100A were consistent with those of the parental strain C58 Δ1027. At the same time, mutation in His100 did not affect the interactions between Atu1027 and two CheWs (Fig. S5). These results indicated that His100 plays a crucial role in the functionality of the globin domain within Atu1027. Additionally, this finding also provides evidence to support the point that Atu1027 acts as an aerotaxis receptor.

Figure 7 Effect of His100 within Atu1027 on chemotaxis and aerotaxis of A. tumefaciens.

The chemotaxis to sucrose and aerotaxis to atmospheric air was tested among wild-type C58, C58Δ1027, and C58Δ1027H100A. (A) The swimming plate containing sucrose as the carbon source shows typical swimming halos of C58, C58Δ1027, and C58Δ1027H100A on the swimming plate. (B) Statistical results of swimming halo diameter from C58, C58Δ1027, and C58Δ1027H100A at 48 h post-inoculation. The different lowercase letters indicate significant differences between groups (P <  0.05).(C) A typical photograph of bacterial aerotaxis in the air trap device at 120 h post-inoculation. Yellow arrows indicate the positions of the bacteria in the Pasteur pipette.

Deletion of atu1027 impaired the biofilm formation of A. tumefaciens in static culture

Biofilm formation is an integral aspect of the interactions between plants and microbes, playing a critical role in the establishment of infections (Bogino et al., 2013; Danhorn & Fuqua, 2007). In the soilborne pathogen Ralstonia solanacearum, the normal formation of biofilms necessitates the involvement of aerotaxis (Yao & Allen, 2007). Atu1027 serves as an aerotaxis receptor to modulate chemotaxis in A. tumefaciens, the deletion of atu1027 may impair the biofilm formation of A. tumefaciens. Hence, the biofilm formation assays of atu1027 mutants were performed in static culture. As shown in Fig. 8, at 72 h post-inoculation, all strains under examination possess the ability to generate biofilms. However, it is noteworthy that the biofilms produced by strains C58 and C58 Δ1027+ exhibit considerably greater strength compared to those formed by C58 Δ1027. This observation serves to validate the notion that the absence of Atu1027 diminished the biofilm formation capacity of A. tumefaciens in static culture.

Figure 8 Biofilm formation of A. tumefaciens C58 and its Atu1027 mutant strains.

The biofilm formation of wild-type C58, C58Δ1027, and C58Δ1027+ was quantified by a crystal violet (CV) staining method. The OD570 value of the 30% acetic acid (vol/vol) -solubilized CV and the OD600 value of the planktonic cells were recorded and calculated. To exclude the influence of the growth rate of different strains on biofilm formation under static conditions, the ratio of OD570/OD600 was used to normalize the differences in culture growth between strains and to quantify the biofilm formation of C58, C58Δ1027, and C58 Δ1027+. The different lowercase letters on the bar indicate significant differences between different groups (P < 0.05).

atu1027 is required for the full pathogenicity of A. tumefaciens

The deletion of the atu1027 gene impairs not only chemotaxis and aerotaxis of A. tumefaciens but also its ability to form biofilm in static culture. Nevertheless, it remains unknown the impact of deletion of atu1027 on the pathogenicity of A. tumefaciens. To investigate the potential effect of atu1027 deficiency on the pathogenicity of A. tumefaciens, three different strains were used: the wild type C58, the atu1027-deficient mutant C58 Δ1027, and the complemented strain C58 Δ1027+. These strains were individually inoculated onto the leaves of Kalanchoe plants. All tested strains were precultured in AB-sucrose liquid medium to mid-log phase, normalized to the same cell density, and inoculated on the Kalanchoe plants’ leaves. Based on the tumour photograph presented in Fig. 9A, it can be observed that, except the avirulent strain C58 ΔvirA, the other three tested strains were capable of inducing tumours on the wound sites of Kalanchoe leaves after 30 days of inoculation. However, the tumour induced by the C58 Δ1027 was smaller compared to those induced by the C58 and C58 Δ1027+. The data presented in Fig. 9B further supports this observation, as it demonstrated that the tumour weight induced by the strain lacking atu1027 was significantly lower than that induced by the other tested strains. Hence, it can thus be concluded that deletion of atu1027 gene attenuates the pathogenicity of A. tumefaciens.

Figure 9 Tumorigenesis assays of A. tumefaciens C58 and its Atu1027 mutant strains.

Different A.  tumefaciens strains were grown to the middle-log phase in AB-sucrose liquid medium. The cell concentration of all tested strains was adjusted to 5 ×108 cfu/mL. 3 µL cell suspension of different tested strains were respectively inoculated onto the wound lines on the leaves of Kalanchoe plants. An avirulent strain C58ΔvirA was used as a negative control. The Kalanchoe plants were grown at room temperature. The tumors on the leaves were photographed 30 days after inoculation. The tumors in each wound line were carefully scraped and weighted to quantify the pathogenicity. Data are the means of three biological replicates with the standard error. The lowercase letters above the bars represent significant differences obtained by one-way analysis of variance, followed by all pairwise multiple comparison procedures (SNK test) (p < 0.05).

Discussion

In the present study, we conducted a characterization of a cytoplasmic chemoreceptor, Atu1027, in A. tumefaciens C58. Atu1027 was found to possess a globin domain and functioned as both an aerotaxis receptor and a participant in the chemotactic response to six carbon sources. The globin domain was identified as the heme-binding domain within globin-coupled sensors (GCS) (Hou et al., 2001a). The GCS proteins are a group of putative O2 sensors that contain an N-terminal sensor domain, a variable middle domain, and a C-terminal output domain (Rivera et al., 2018; Walker, Rivera & Weinert, 2017). Oxygen-dependent physiology and phenotypes in microorganisms are likely controlled by GCS proteins. Based on the C-terminal output domains, the GCS proteins can be categorized into various types, including MCP, kinase, diguanylate cyclase (DGC), and adenylate cyclase (AC) (Walker, Rivera & Weinert, 2017). Previous research has identified MCP-type GCS proteins in the archaeon H. salinarium (Hou et al., 2000), as well as in two gram-positive bacteria, B. subtilis and Bacillus halodurans (Hou et al., 2000; Hou et al., 2001b). Our study provided evidence that Atu1027 functions as a typical MCP-type GCS protein and serves as an aerotaxis receptor in the gram-negative bacteria A. tumefaciens. The heme-binding site (His100) located within the globin domain of Atu1027 plays a critical role in facilitating the chemotaxis and aerotaxis functions of Atu1027. More importantly, the amino acid sequence of Atu1027 is highly conserved among the chemoreceptors in different Agrobacterium species (Fig. S2). This observation suggests that MCP-type GCS proteins are widely present in Agrobacterium, potentially serving as the main aerotaxis receptor to facilitate multiple physiological activities.

Moreover, it should be noted that Atu1027 is not the sole predicted chemoreceptor that possesses a globin domain. The atu0514 gene, located within the chemotaxis operon of A. tumefaciens C58, also encodes a chemoreceptor that contains a globin domain (Xu et al., 2020). Atu0514 affected the chemotactic response of A. tumefaciens to various chemoeffectors, but its function in aerotaxis has not been addressed (Ye et al., 2021). To further investigate, we conducted an alignment of the globin domain of Atu0514 with that of HemAT-Bs, HemAT-Hs, and Atu1027. Figure S6 illustrated that three conserved amino acids, which are present in all globins, were not identified within the globin domain of Atu0514. Therefore, it is hypothesized that Atu0514 may possess unique characteristics that distinguish it from typical globin domain-type chemoreceptors. Given its exclusive presence as the sole chemoreceptor encoded by the chemotaxis operon of A. tumefaciens, Atu0514 likely plays a crucial role in the transduction of chemotaxis signaling.

In addition to the globin domain, the PAS domain is another common sensor mediating aerotaxis occurring in chemoreceptors. However, the specific functions of the PAS domain in chemoreceptors exhibit variability. The involvement of the PAS domain in aerotaxis primarily encompasses two types of mechanisms. Type I involves the binding of a redox-sensitive flavin adenine dinucleotide (FAD) cofactor by the PAS domain to monitor changes in the redox status of the electron transport system. Examples of this type include Aer in E.coli (Bibikov et al., 2000; Maschmann et al., 2022) and AerC in Azospirillum brasilense (Xie et al., 2010). The second mechanism, known as Type II, involves the PAS-containing chemoreceptor binding to a heme cofactor. This binding enables the chemoreceptor to sense oxygen and regulate aerotaxis, two typical examples of Type II mechanism are IcpB in Azorhizobium caulinodans (Jiang et al., 2016) and Aer2 in Pseudomonas aeruginosa (Garcia et al., 2017). Interestingly, type II is comparable to that observed in the B. subtilis, but the heme cofactor was bonded by a globin-coupled domain rather than a PAS domain (Hou et al., 2000). In the case of A. tumefaciens, four putative chemoreceptors contain two PAS domains within their N-terminal region, yet the precise function of these chemoreceptors remains unknown (Xu et al., 2020). Based on our study findings, we propose that Atu1027 serves as an aerotaxis receptor. Both qualitative assays used in this study showed that Atu1027 deletion prevents aerotaxis.

It is reported that chemotaxis and aerotaxis are required for normal biofilm formation and interaction with the hosts (Arrebola & Cazorla, 2020; Greer-Phillips, Stephens & Alexandre, 2004; Hölscher et al., 2015; Ichinose et al., 2023; Jiang et al., 2016; Tumewu et al., 2022; Yao & Allen, 2007). Similar findings were observed in the investigation of Atu1027, where the removal of atu1027 resulted in a modest reduction in biofilm formation and pathogenicity of A. tumefaciens. One potential explanation for the disparity observed in biofilm formation between the atu1027-deficient mutant and the wild-type C58 is the impaired capacity of atu1027-deficient agrobacterial cells in response to the air, resulting in a reduced population of bacterial cells at the liquid-air interface (Fig. 5). This scarcity of bacterial cells close to the solid–liquid interface could potentially hinder the initial growth of the biofilm. Within further growth and maturation of the biofilm, the local environment within the biofilm undergoes several changes, notably a decrease in the availability of oxygen (Stewart & Franklin, 2008). Wild-type C58 with aerotaxis is more competitive when compared with C58 Δ1027 without aerotaxis. It is worth noting that His100 is crucial for Atu1027 to sense oxygen, we found that there is no difference in the biofilm formation between C58 Δ1027H100A and C58 Δ1027 (Fig. S7), which also suggested the contribution of aerotaxis to biofilm formation. For the pathogenicity, when A. tumefaciens infects plants and forms crown gall tumors on the wound sites, the oxygen concentration in the crown gall tumors is also lower than that on the outside (Gohlke & Deeken, 2014). Wild-type C58 carrying Atu1027 has an advantage in finding the appropriate oxygen concentration over the atu1027-deficient mutant. An adequate oxygen supply is more conducive to active physiological activity and promotes the enhancement of the pathogenicity of A. tumefaciens. More importantly, compared with the biofilm formed by C58 Δ1027, the stronger biofilm formed by wild-type C58 also helps the bacteria to establish a more effective infection and against harsh environmental factors.

However, to our knowledge, the involvement of aerotaxis receptors in the process of biofilm formation is controversial. Some of the aerotaxis receptors act as a negative regulator in biofilm formation (Jiang et al., 2016; Yao & Allen, 2007), while others serve as positive regulators (Arrebola & Cazorla, 2020; Hölscher et al., 2015; Tumewu et al., 2022). Our results were in according with the latter. To the best of our knowledge, there is still no literature that completely explains how the aerotaxis receptor affects biofilm formation in microorganisms. In our opinion, subsequent in-depth studies need to be done to explain the molecular mechanisms of aerotaxis receptors in regulating biofilm formation. The investigation of cross-talk between the chemotaxis signal transduction pathway and the biofilm formation represents a promising avenue for further research. A recent study has found that some chemoreceptors in Comamonas testosteroni CNB-1 can physically bind with the components in the pathway of biofilm formation (Huang et al., 2019). Additionally, the histidine kinase CheA has been observed to phosphorylate FlmD, a response regulator that plays a crucial role in mediating biofilm formation (Huang et al., 2019). Furthermore, the role of microbial species and their energy metabolism types cannot be ignored.

Conclusions

Atu1027, a cytoplasmic chemoreceptor found in A. tumefaciens C58, exhibits an N-terminal globin domain and serves as an aerotaxis receptor, governing the organism’s response to atmospheric air as well as its chemotactic movement towards diverse carbon sources. The conserved residue His100 within Atu1027 plays a pivotal role in facilitating its globin domain’s functionality in both aerotaxis and chemotaxis. Furthermore, Atu1027 exerts influence over agrobacterial chemotaxis and the invasion of A. tumefaciens into its host.

Supplemental Information

Supplemental Information 1 Strategy for constructing atu1027-deficient mutant

(A) the schematic depicting the process of PCR splicing through overlapping extension and the subsequent construction of the suicide plasmid pEX-d1027 is presented. Specifically, a 585 bp segment upstream and a 587 bp segment downstream of the atu1027 ORF were spliced together using overlapping PCR. These spliced segments were then inserted into the pEX18Km plasmid using the BamH I and EcoR I restriction enzymes. The resulting suicide plasmid, pEX-d1027, contained both the upstream (indicated by purple arrows) and downstream (indicated by brilliant blue arrows) sequences of the atu1027 gene. This plasmid was subsequently introduced into agrobacterial cells and integrated into the genome DNA at the targeted gene locus through homologous recombination. The sacB gene, indicated by yellow arrows, serves as a counterselectable marker gene, conferring sensitivity to sucrose. The nptIII gene, represented by gray rectangles, confers resistance to kanamycin. Colonies that have undergone single cross-over homologous recombination were screened for kanamycin resistance and sucrose sensitivity. The second cross-over homologous recombination event was identified by plating the single cross-over colonies on media containing 5% sucrose. The colonies that exhibit growth solely on MG/L media containing 5% sucrose, but not on MG/L media containing 100 µg/mL, can be utilized for identification of atu1027-deficient mutants. There are two potential strategies that can occur within cells. (A) the first strategy involves the utilization of the upstream sequence of atu1027 in a gene replacement approach. (B) the second strategy involves the utilization of the downstream sequence of atu1027 in a gene replacement approach.

Click here for additional data file.

Supplemental Information 2 Amino acid sequence alignment of globin domains in Agrobacterium species

The globin domain of Atu1027 was aligned with homologous proteins including WP_269710637.1 from Agrobacterium rhizogenes, WP_269833196.1from Agrobacterium salinitolerans, MCZ7463360.1 from Agrobacterium rhizogenes, WP_080856168.1 from Agrobacterium deltaense, WP_207132785.1 from Agrobacterium burrii, WP_269827917.1 from Agrobacterium leguminum, WP_077987755.1 from Agrobacterium pusense.

Click here for additional data file.

Supplemental Information 3 The original western blot image of Fig. 3C

Click here for additional data file.

Supplemental Information 4 Growth curves of A. tumefaciens wild-type and atu1027 mutant strains

A. tumefaciens wild-type and atu1027 mutant strains were inoculated in 5 mL of MG/L medium and cultured for 12 h. Then all the tested strains were washed and re-suspended in fresh AB-sucrose medium. The cell density of different suspensions was adjusted to the same. After that, one mL of cell suspension was added to 100 mL of AB-sucrose liquid medium to determine growth curves.

Click here for additional data file.

Supplemental Information 5 The interactions between Atu1027H100A and two CheWs

Click here for additional data file.

Supplemental Information 6 Amino acid sequence alignment of globin sensor domain of HemAT-Bs (B. subtilis), HemAT-Hs (H. salinarum), Atu1027 (A. tumefaciens) and Atu0514 (A. tumefaciens)

Sequences of four proteins were full aligned using the Clustal program of the DNAMAN 8. All settings used default values. In the figure on the right, the numbers indicate the position of the amino acid. Asterisks represent residues that are absolutely conserved in all globins. Dark spots represent residues that are highly conserved (Bashford, Chothia & Lesk, 1987; Vinogradov, Walz & Pohajdak, 1992; Hou et al., 2000).

Click here for additional data file.

Supplemental Information 7 Biofilm formation assay of wild-type C58 and Atu1027 mutants

The quantification of biofilm formation in wild-type C58, C58 Δ1027, and C58Δ1027H100A was conducted using the crystal violet (CV) staining method. The OD570 value of the CV solubilized in 30% acetic acid (vol/vol) and the OD600 value of the planktonic cells were measured and computed. Distinct letters on the bar represent statistically significant differences between the various groups (P < 0.05).

Click here for additional data file.

Supplemental Information 8 Bacterial strain and plasmids used in this study

Kmr = Resistant to kanamycin; Surs = Sensitive to sucrose; Apr = Resistant to ampicillin; Crr = Resistant to carbenicillin; Cmr = Resistant to chloramphenicol; Tcr = Resistant to tetracycline

Click here for additional data file.

Supplemental Information 9 Primers used in the study

a Primers in the brackets indicate that these primers can amplify more than one sequence for different purposes.

b Restriction enzyme sites are highlighted with underlines.

Click here for additional data file.

Supplemental Information 10 Information about the sequences used in the consensus phylogenetic tree

Click here for additional data file.

Supplemental Information 11 Wild-type C58 cells response to atmospheric air

Click here for additional data file.

Supplemental Information 12 atu1027-deficient mutant C58Δ1,027 cells response to atmospheric air

Click here for additional data file.

Supplemental Information 13 The complemented strain C58Δ1027+ cells response to atmospheric air

Click here for additional data file.

Supplemental Information 14 Raw data and images

Click here for additional data file.

We thank Qingxuan Zhou (College of Bioscience and Biotechnology, Yangzhou University) for the mutant construction. We thank Prof. Xiangqian Li, Dr. Shiyan Wang, Dr. Guodong Chen, and Yi Jiang (School of Life Science and Food Engineering, Huaiyin Institute of Technology) for their support in providing experimental equipment.

Additional Information and Declarations

Competing Interests

Author Contributions

Data Availability

The authors declare there are no competing interests.

Zhiwei Huang conceived and designed the experiments, performed the experiments, analyzed the data, prepared figures and/or tables, authored or reviewed drafts of the article, and approved the final draft.

Junnan Zou conceived and designed the experiments, performed the experiments, analyzed the data, prepared figures and/or tables, authored or reviewed drafts of the article, and approved the final draft.

Minliang Guo conceived and designed the experiments, analyzed the data, authored or reviewed drafts of the article, and approved the final draft.

Guoliang Zhang conceived and designed the experiments, authored or reviewed drafts of the article, and approved the final draft.

Jun Gao analyzed the data, authored or reviewed drafts of the article, and approved the final draft.

Hongliang Zhao analyzed the data, authored or reviewed drafts of the article, and approved the final draft.

Feiyu Yan performed the experiments, prepared figures and/or tables, and approved the final draft.

Yuan Niu performed the experiments, prepared figures and/or tables, and approved the final draft.

Guang-Long Wang performed the experiments, analyzed the data, authored or reviewed drafts of the article, and approved the final draft.

The following information was supplied regarding data availability:

The raw measurements are available in the Supplementary File.

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
