# Peer review of "An aerotaxis receptor influences invasion of Agrobacterium tumefaciens into its host"

_PeerJ, doi:10.7717/peerj.16898_

## Round 0.1 · original submission · Major Revisions

Two experts assessed your manuscript and found the content relevant. However, some of the experimental approaches have room for improvement, and as a consequence, the results currently provide inconclusive information.

·

Basic reporting

The manuscript by Zhi-Wei Huang and colleagues presents the potential role of Atu1027 gene as an oxygen and chemotactic response in A. tumefaciens. Overall, the manuscript is professionally written and contains interesting and relevant data. However, this reviewer feels some points need clarification and further experimental support. In the following sections, I provide some comments in good faith to strengthen the manuscript.

Experimental design

Overall, the experimental design is sound, but some comments are provided.
The statistical test used agrees with the experimental approaches and is sufficient for showing the differences between treatments.

Validity of the findings

The report provides interesting information, but there are some aspects that this reviewer finds important to be addressed. In the following lines, I comment on the issues this reviewer thinks need to be clarified in the manuscript or to me.
Some important material missing in the paper is the deletion strategy as a diagram, showing the steps and the plasmid maps for each step. Also, in the strategy, the primers used should be indicated to highlight the regions amplified and the purpose of each fragment. I understand that the authors cite the proper papers, but for the reader is not easy to follow. Please concentrate all the required data to follow the constructs, their purpose and the steps followed in a single diagram; this can be provided as Figure 2 or as supplementary material.
In the methods, please include the concentration used for each carbon source in the chemotaxis assay.
The results shown in Figure 2 for this reviewer are inconclusive. The two-hybrid assay in both the qualitative and the quantitative assays (Figure 2A and 2B) has a slightly higher activity in one of the three clones shown. Also, although statistically significant, the beta-galactosidase assay is not enough for this reviewer to assess the true interaction. I strongly suggest devising a protein-protein interaction in vitro, using recombinant proteins, to show that Atu1027 interacts with both CheW proteins unambiguously. One thing that this reviewer has found useful in these assays is to place 3 µL droplets into the agar plate instead of plating the colonies, as shown here. The more homogeneous the colony is, the more consistent the analysis is and helps visually inspect the result. Please substitute the figure with a plate as suggested and try to provide a strong case for the result shown here.
For this reviewer, the images in Figure 3 panel B are unclear. I suggest using a better zoom-in of each image and indicating the cell's position with an arrow instead of the yellow squares. Also, for the microscopic imaging, I recommend a video showing the bacterial response in the supplementary material. The figure should include in all panels the borderline for the drop edge.
For this reviewer, the least convincing result is shown in Figure 5. First, the swimming halo is not the same as a colony. Here, the authors should refer to the swimming halo. Second, the reduction of the swimming halo is not dramatic. Thus, authors could interpret this as Atu1027 belonging to the chemotactic complex but perhaps not directly involved in flagella control. As a reviewer, I need more convincing evidence that Atu1027 is directly engaged in CheW interaction and, thus, flagella control.
Regarding the H100A mutant, this reviewer thinks that authors should provide evidence that the protein is not aggregating and suffering some kind of overall malfunction unrelated to the unique mutation introduced.
Authors are invited to include the H100A mutant in the biofilm formation assay to show the complete set.
Figure 8 panel A is too small; please enlarge this image.
Regarding the distribution of Atu1027, this reviewer suggests including a broad search of homologous proteins and providing a phylogenetic reconstruction to assess the evolutionary relationship with Aer, for example, and homologous proteins and reinforce the role of Atu1027 in oxygen sensing.
Please modify accordingly if the additional information adds to the discussion and conclusions.

Additional comments

Please correct the writing in line 54, “microbes” is repeated here.
In line 125 please add heme cofactor.
In Table S1 please add the alpha symbol in DH5alpha.
In Table S2 please add the Tm for each primer, target amplification and primer length.
In line 171, the Inoue protocol should be cited and not the Sambrook reference alone; perhaps one reference is for the plasmid isolation and the Inoue protocol for transforming E. coli cells. Please revise.
In line 205 please correct to coli.
In line 250, please change to “a single carbon source”.
In line 262 please place as subindex the 600 in OD600.
In the methods section, please provide the centrifugation conditions expressed in xg.
In line 332, please complete the phrase. “Consequently, the results indicated that…” the idea is incomplete.
In lines 401-403, I suggest adding in the Figure 1 panel A the position of the key residues mentioned in these lines.
Lines 503-504 seems incomplete. Which qualitative assays are the authors referring to? I think it should read, “The qualitative assays used in this study, showed that…”
In line 531 I think is better to state “to our knowledge” instead of “current studies”.
Please include the accession number of Atu1027.
Other minor comments include: please describe the settings and type of alignment used in this study.

Reviewer 2 ·

Basic reporting

See "additional comments" section.

Experimental design

See "additional comments" section.

Validity of the findings

See "additional comments" section.

Additional comments

This study investigates the role of the chemoreceptor Atu1027, with a globin-like sensor domain, in the aerotactic properties of the phytopathogen Agrobacterium tumefacients. The authors observed that a mutant in the atu1027 gene does not exhibit aerotaxis and is deficient in biofilm formation as well as phytopathogenicity. In general, the experiments are well conducted and the results are coherent. Some deficiencies in the English style as well as some conceptual errors - especially in the introduction - were noted.
Main comments:
- The section of the introduction that describe the functioning of the signal transduction pathway involved in chemotaxis is a bit confusing and distracting, and could be explained in a more orderly and simplified manner. In this section, there are parts that are redundant with respect to previous descriptions (e.g., lines 92-93). Also, the classification of chemoeffectors into types I and II (lines 94-99) seems highly artificial. For example pH, classified as part of the chemoeffectors in the type II category, could not be considered a chemical compound, as the authors do.
- Lines 125-127: The authors state that there are three types of ligand binding domains (LBD) of chemoreceptors that respond to aerotaxis. This does not seem to be entirely correct - especially in the case of 4HB-type LBDs (the LBD present in the Tsr chemoreceptor). The authors should clarify that the mechanisms by which Tsr/Tsr-like chemoreceptors respond to aerotaxis/energy taxis seem to be dependent on charged amino acids in the HAMP domain and not to LBD sensing.
- Lines 247-258 & Lines 384-398: the authors must distinguish between chemotaxis, aerotaxis and energy taxis to a greater extent. Thus, they should justify why the deletion of the chemoreceptor gene atu1027 may affect chemotactic responses (other than aerotaxis). Previous data revealed that Aer and Tsr receptors mediate taxis to metabolizable compounds via energy taxis, which responds to internal energetic conditions and not to changes in the chemical gradients of the environment. In addition, to investigate chemotaxis towards metabolizable compounds of the wild-type and atu1027 mutant strain, the authors performed plate motility assays. These assays do not distinguish between chemotaxis and energy taxis, so the results obtained do not allow to draw conclusions - mainly when the radii/diameters of the images provided in Fig. 5A are extremely similar between the parental and mutant strain (the values in histograms 5B and those of the images in Fig. 5A do not seem to match). This referee considers that authors should complement their swimming plate assays with capillary chemotaxis assays.
- Tables S1 and S2 seem incomplete and must be revised. For example, there is no information about plasmids and oligonucleotides used for the generation of the virA mutant strain. There is not description of the oligonucleotides used to construct the cheW1 and cheW2 plasmids for the two-hybrid assays, etc.
- In the discussion section, the authors discuss about the possible role/function of two chemoreceptors, one with another globin type ligand binding domain and the other with a PAS domain. The authors could propose a function for the latter receptor based on the recently published paper.
Specific comments:
- Line 46: “Agrobacterium tumefaciens (also called Agrobacterium fabrum)”. The name of two species cannot be synonymous. It should read "reclassified as…". Also, has A. tumefaciens been reclassified as “Rhizobium radiobacter”?
- For clarity, to distinguish between the MCP signaling domain and the receptor protein, the authors should use in the text the term "chemoreceptor" and not "MCP/methyl-accepting chemotaxis protein".
- Lines 133-149: the authors should describe the type of LBD that the characterized chemoreceptors of A. tumefaciens possess.
- Lines 145-146: “…ligands of Atu1912 are propionate and propionic acid”. Aren't propionate and propionic acid the same compounds in acid and salt form?
- Lines 159, 162, etc: the authors should leave a space between the number and the units.
- Some sections of the materials and methods could be reduced in length. Also, the section of the materials and methods explaining the assembly and functioning of the air trap is confusing and should be re-writtend. This section should also refer to the schematic in Figure 3A.
- Line 262: 600 nm should be as subindex.
- Line 278: should read “assays described previously” instead of “assays described”.
- Line 301: all active chemoreceptors contain a HAMP and a signaling domain so the information provided here is not relevant.
- Line 332: incomplete sentence.
- Line 344-347: redundant information. Already described in the experimental procedures section.
- Lines 318-319 & 380 & 381: these sentences are redundant. Check redundancies throughout the text.
- Line 434: change atu1027 to Atu1027, if the authors refer to the chemoreceptor; or to atu1027 (in italics) if they intend to refer to the gene.
- Line 441: A. tumefaciens must be in italics.
- Lines 436-454: the authors do not mention anything about the non-virulent phenotype of the virA mutant strain in the plant assays shown in Fig. 8.
- Lines 465-467: reference needed after “… and adenylate cyclase (AC)”.
- Line 469: “Bacillus halodurans C-125”. Why do authors specify the name of the strain and in other cases only the name of the species (e.g. B. subtilis)?
- Growth (OD660 nm) in Fig. S2 should be given in logarithmic scale.
- Legend of Figure 7: “Biofilm formation of A. tumefaciens C58 and its Atu1027 mutant strains”. There is only a mutant strain in this figure.
- Table S: “R” in CmR (in pBT-LGF2) and TcR (pTRG-GA111) must be in superindex. atu1027 in pBT-1027 must be in italics.
- In general, the English style of the manuscript could be improved.

---

## Round 0.2 · Minor Revisions

The manuscript still needs improvement in the bioinformatics analyses and results explanations, as recommended by Reviewer 1. In addition, supplementary images to validate western blots are required.

·

Basic reporting

Dear authors,
Thank you for addressing most of the comments that I made based on the first version of the manuscript.
Still, there are some issues that as a reviewer I still think they need clarification in the manuscript, please find them in the “Validity of findings” section.

Experimental design

No comment in this section

Validity of the findings

First, the raw data for the CheW western blot needs the Ponceau red to unambiguously show that the protein loaded is the same in the two lanes.
In the response letter point 8, I disagree with the authors that the H100A mutant biofilm and pathogenic experiments will interfere with the reviewer’s understanding of the research objectives. I think that the results shown here are relevant and the complete set should be carried out. If the authors indeed did the experiments and the results are inconsistent, there must be some explanation and perhaps a model can be proposed. Please address these experiments and provided a justification other than the understanding of the objectives.
This reviewer partially agrees with the authors in response letter point 10. The authors could consider HMMER (Hidden Markov Models) for assessing the homologs and reconstruct the evolutionary history of Atu1027. This reviewer thinks that the protein is very interesting and providing the basis for further research in other biological models will give the manuscript more citations. The Figure 1B indeed shows that it has homology to oxygen sensing but is not providing the extent of this protein in other organisms.
Nevertheless, the manuscript was improved greatly and this reviewer thanks the authors for their interesting paper.
All the best for 2024.

Additional comments

No comment in this section.

---

## Round 0.3 · accepted · Accept

The authors addressed the concerns raised in the previous revision round.